# Blackberry-Loaded AgNPs Attenuate Hepatic Ischemia/Reperfusion Injury via PI3K/Akt/mTOR Pathway

**DOI:** 10.3390/metabo13030419

**Published:** 2023-03-13

**Authors:** Ahmed M. Fathi, Shaimaa Waz, Eman Alaaeldin, Nisreen D. M. Toni, Azza A. K. El-Sheikh, Ahmed M. Sayed, Usama Ramadan Abdelmohsen, Maiiada H. Nazmy

**Affiliations:** 1Department of Biochemistry, Faculty of Pharmacy, Minia University, Minia 61519, Egypt; 2Department of Pharmaceutics, Faculty of Pharmacy, Minia University, Minia 61519, Egypt; 3Department of Pharmaceutics, Faculty of Pharmacy, Deraya University, New Minia 61111, Egypt; 4Department of Pathology, Faculty of Medicine, Minia University, Minia 61519, Egypt; 5Basic Health Sciences Department, College of Medicine, Princess Nourah Bint Abdulrahman University, P.O. Box 84428, Riyadh 11671, Saudi Arabia; 6Department of Pharmacognosy, Faculty of Pharmacy, Nahda University, Beni-Suef 62513, Egypt; 7Department of Pharmacognosy, Faculty of Pharmacy, Minia University, Minia 61519, Egypt; 8Department of Pharmacognosy, Faculty of Pharmacy, Deraya University, New Minia 61111, Egypt

**Keywords:** liver ischemia-reperfusion injury, blackberry, caspase-3, PI3K/Akt/mTOR, AgNPs, cell apoptosis, metabolomics

## Abstract

Liver ischemia-reperfusion injury (IRI) is a pathophysiological insult that often occurs during liver surgery. Blackberry leaves are known for their anti-inflammatory and antioxidant activities. Aims: To achieve site-specific delivery of blackberry leaves extract (*BBE*) loaded AgNPs to the hepatocyte in IRI and to verify possible molecular mechanisms. Methods: IRI was induced in male Wister rats. Liver injury, hepatic histology, oxidative stress markers, hepatic expression of apoptosis-related proteins were evaluated. Non-targeted metabolomics for chemical characterization of blackberry leaves extract was performed. Key findings: Pre-treatment with *BBE* protected against the deterioration caused by I/R, depicted by a significant improvement of liver functions and structure, as well as reduction of oxidative stress with a concomitant increase in antioxidants. Additionally, *BBE* promoted phosphorylation of antiapoptotic proteins; PI3K, Akt and mTOR, while apoptotic proteins; Bax, Casp-9 and cleaved Casp-3 expressions were decreased. LC-HRMS-based metabolomics identified a range of metabolites, mainly flavonoids and anthocyanins. Upon comprehensive virtual screening and molecular dynamics simulation, the major annotated anthocyanins, cyanidin and pelargonidin glucosides, were suggested to act as PLA2 inhibitors. Significance: *BBE* can ameliorate hepatic IRI augmented by *BBE*-AgNPs nano-formulation via suppressing, oxidative stress and apoptosis as well as stimulation of PI3K/Akt/mTOR signaling pathway.

## 1. Introduction

Liver ischemia-reperfusion injury (IRI) occurs when the blood flow interrupted following liver surgery which is accompanied by restoration. Liver IRI is also a significant complication of hepatic tumor resection, transplantation, hypovolemic shock, and other hepatic surgeries [1]. Liver I/R causes hepatocyte necrosis and results in structural damage to the liver [2,3]. Several mechanisms have been proposed to describe the pathogenesis of liver I/R injury including adenosine triphosphate (ATP) depletion, reactive oxygen species (ROS) overproduction, macrophage activation, inflammation, and apoptosis [4,5].

Liver I/R injury consists of complex processes [6]. Adenosine triphosphate (ATP) depletion interferes with metabolic and transport processes dependent on cellular energy [6,7]. The mechanism of reperfusion consists of two phases: in the initial step, activated resident liver macrophages and Kupffer cells, induce oxidative stress primarily through the production of ROS and, in the later phase, recruited neutrophils release inflammatory mediators capable of causing cell apoptosis or death [8,9]. Specific morphological characteristics and energy-dependent biochemical pathways have been correlated with programmed cell death or apoptosis [10], however upstream activators like Caspase cascade signaling, Bax pro-apoptotic proteins, cellular stress triggers, and hypoxia have often induced apoptotic pathways [11].

PI3K/Akt/mTOR signaling promotes cell survival, is involved in brain, kidney, and hepatic ischemia/reperfusion injury, and is linked to apoptosis and inflammation [12,13,14]. When activated, PI3K sends extracellular signals to its downstream Akt, which promotes cell survival by regulating apoptosis-triggering proteins. The inhibition of Bax by Akt reduces cell apoptosis by suppressing caspase-3 activity [15,16]. PI3K/Akt/mTOR signaling is a typical anti-apoptotic signaling pathway and its involvement in the process of hepatic IRI has been documented [17,18,19]. It has been reported that the pro-apoptotic proteins including Bax, Caspase-9 and Caspase-3 mediate mitochondrial apoptosis which occur in hepatic IRI [20,21]. Therefore, inhibition of cell apoptosis in the management of I/R injury can therefore be of clinical importance [22].

Plant natural products have been widely used as hepato-protective. The blackberry leaves were used for many medicinal purposes and are considered a great source of natural antioxidants. Recent studies have shown that blackberry extract is capable of reducing the risk of hepatic disease and cancer, as well as exhibiting a high antioxidant capacity [23,24,25,26] Blackberries have been shown to yield higher levels of anthocyanins as well as other antioxidants, including flavonols, phenolic acids, ellagic acid, vitamins C and E [27]. Among these components, the anthocyanins, which are the main polyphenolic compounds in blackberries, are outstanding scavengers of a broad variety of ROS, such as superoxide radical, hydrogen peroxide, and hydroxyl radical [28]. Polyphenolics from blackberries have been reported to exhibit antioxidant and anti-inflammatory activities [29]. In addition, many in vitro and in vivo studies have recorded that blackberry anthocyanins, as antioxidants and/or anti-inflammatory agents, have potential biological benefits [30,31], in addition, it was reported that anthocyanins has an ameliorative effect on cerebral and cardiac IRI [32,33]. Cyanidin-3-glucoside from blackberries has been observed to show chemo-preventive and chemotherapeutic efficacy in vivo and in vitro [34]. It was observed that Cyanidin-3-glucoside has a protective effect on cardiac and hepatic IRI [35,36]. However, the in vivo antioxidant function and mechanisms of blackberry extract have not been well understood.

However, employing plant extracts alone to cure various diseases is either useless or slow, and they require more and more enhancement to show their effectiveness [37]. Nanotechnology and its implementations in science and technology have recently made significant progress. Metallic nanoparticles such as silver nanoparticles (AgNPs) have presented significant issues in health and medicine [38]. Earlier, several studies investigated the use of various plant extracts and natural products to bind with silver as nanocarriers to be used as anti-cancer, anti-neurotoxicity, anti-diabetic, and nephroprotective agents [37].

Nanoparticles were created using a solid-state process. The benefit of utilizing such a technique is that it saves time, chemicals, and solvents that can be utilized extensively in traditional synthesis, as well as ease of transportation from the lab to various industrial domains [39,40]. The aim of this work is to study the impact of AgNPs in magnifying the benefits of blackberries in mitigating hepatic IRI depending on its antioxidant and hepatic injury-attenuating properties.

## 2. Materials and Methods

### 2.1. Materials

Full-ripe blackberry leaves were acquired from “Egyptian local market”. AgNO_3_ powder was purchased from “Merck Co. Germany”. Pectin and sodium hydroxide pellet (NaOH) were purchased from Adwic Company, (Cairo, Egypt). Silymarin was delivered from Medical Union Pharmaceuticals, (Ismailia, Egypt). All other chemicals were used exactly as they were delivered. For preparation, characterization, and in-vivo experiments, deionized water was used.

### 2.2. Kits and Antibodies

Alanine transaminase (ALT), Aspartate transaminase (AST), malondialdehyde (MDA), reduced glutathione (GSH), Catalase (CAT) and Superoxide dismutase (SOD) colorimetric kits were obtained from Biodiagnostic Laboratory Reagents (Cairo, Egypt). Total protein colorimetric kit was obtained from Thermo Fisher Scientific, (Waltham, MA, USA) with catalog numbers (23225). Polyclonal rabbit hosted Cleaved Caspase-3, Akt, phosphorylated (p)-Akt antibodies were obtained from Elabscience Biotechnology Inc. (Wuhan, Hubei, P.R.C., China) with catalog numbers (E-AB-30004; E-AB-30465; E-AB-20802) respectively. Phosphorylated (p)-PI3K, (p)-mTOR and β-Actin antibodies were obtained from Santa Cruz Biotechnology (Dallas, TX, USA) with catalog numbers (sc-1637; sc-293133; sc-47778) respectively. Bax and cleaved caspase-9 antibodies were procured from Abcam (Cambridge, UK) with catalog number (ab32503; ab210611). Biotin- conjugated goat anti-rabbit IgG secondary antibody was procured Invitrogen (Carlsbad, CA, USA) with catalog number (#A16100). Alkaline phosphatase-conjugated goat anti-rabbit and alkaline phosphatase conjugated goat anti-mouse secondary antibodies were procured from Sigma-Aldrich Chemical Company (St. Louis, MO, USA) with catalog numbers (A3687 and A3562) respectively.

### 2.3. Preparation of Blackberry Extract

Blackberry leaves were collected on Jan. 2020 from El-Orman Botanical Garden (Giza, Egypt). Plant was kindly identified by Eng. Trease Labib (former-Head of El-Orman Botanical Garden) and Prof. Nasser Barakat (Faculty of Science, Minia University, Egypt). A voucher specimen (2020-Der 05) was deposited at the Department of Pharmacognosy (Faculty of Pharmacy, Deraya University, New Minia, Egypt). Blackberry leaves were gathered and cleaned, rinsed with deionized water, and dried for four days before being ground thoroughly. The resulting powder (10 g) was thoroughly mixed with absolute ethanol (99%; 50 mL). The supernatants were taken and rotary evaporated at 40 °C. The dried extract was used for dissolving in deionized water, filtered through a 25 μm filter paper, and lyophilized to get dried active constituents. The dried active ingredient was then re-dissolved in deionized water (200 mg/mL), and kept at −20 °C for further characterization. The resulting blackberry extract solution was termed *BBE* [41].

### 2.4. LC–HR–ESI–MS Metabolic Profiling of Blackberry Leaves Extract

Metabolomics analysis using LC-HR-ESI-MS were carried out as described by Abdelmohsen et al. [42]. In brief, plant samples (1 mg/mL in methanol) was analyzed using an Accela HPLC (Thermo Fisher, Dreieich, Germany) equipped with an ACE C18, 75 mm 3.0 mm, 5 m column (Hichrom Limited, Berkshire, UK) and an Exactive (Orbitrap) mass spectrometer (Thermo Fisher). The gradient elution technique was applied at 300 L/min with purified water [TOC was 20 ppb] and acetonitrile, each containing 0.1% formic acid. The elution started with 10% acetonitrile and was gradually increased to 100% acetonitrile within 30 min, followed by an isocratic period of 5 min before returning to 10% acetonitrile for 1 min. The injection volume was 10 μL, and the column temperature was set at 20 °C. With a spray voltage of 4.5 kV and a capillary temperature of 320 °C, HR-ESI-MS was available in both negative and positive ionization modes. Using the insource collision-induced dissociation process, the ESI-MS mass range was assigned at *m*/*z* 100–2000 and *m*/*z* 50–1000. The raw data were imported and analyzed in MZmine 2.12, 27 for differentiation of the HR-MS data, as previously described in detail in Abdelmohsen et al. [42]. A chemotaxonomic filter was applied to the resulting hits to reduce the number of identities per metabolite and include only those that were relevant. As a result, 13 metabolites (1–13) were described using Dictionary of Natural Products and METLIN databases [43].

### 2.5. Solid State Synthesis of AgNPs

AgNPs were prepared adopting the solid state technique. Briefly, pre-determined grinded pectin, stabilizer and reducing agent, was mixed with 0.02 g NaOH (Table 1). After further grinding of the mixture, silver nitrate (AgNO_3_) was added and grinding was maintained for another 3 min. The formation of AgNPs was confirmed when the color of the mixture was changed from colorless to yellowish color. Formed mixture was kept under continuous grinding for 10 min till the color was changed to deep reddish brown. The final product of AgNPs was kept at room temperature for further characterization and application [37].

### 2.6. Preparation of Blackberry Silver Nitrate Nanoparticles (BBE-AgNPs)

To prepare blackberry silver nitrate nanoparticles (*BBE*-AgNPs), 0.2 g of the formed AgNPs was dissolved in 160 mL of deionized water. To ensure complete dissolution of the mixture, magnetic stirring then sonication was maintained for 20 min. Then, *BBE* (40 mL; 10 mg/mL) was added gradually to the prepared solution with stirring for another 30 min. prepared *BBE*-AgNPs were kept for characterization prior to administration to rats [37].

### 2.7. Characterization of the Prepared Metal Nanoparticles

To gain more consent on the formation of AgNPs, spectrophotometric scanning, using UV-vis spectrophotometer (Spectronic Genesys^®^, with Winspec Software, Spectronic, (Pittsford, NY, USA) was carried out. Briefly, a diluted sample (0.001 g/20 mL) was scanned at a wavelength ranging from 250 to 700 nm. In addition, “Transmission electron microscopy”; “TEM; JEOL JEM-2100” operating at 200 kV was utilized for imaging of the prepared *BBE*-AgNPs. Particle size and zeta potential of diluted *BBE*-AgNPs (0.001 g/20 mL) were determined by a “ZETASIZER Nano series; Nano ZS Malvern Instrument Ltd., Malvern, UK” [44].

### 2.8. Entrapment Efficiency

Entrapment efficiency of prepared silver nanoparticles was determined indirectly as described by Hussein, Jihan, et al. [37] Briefly, 1 mL of the prepared solution of *BBE*-AgNPs was suspended in 10 mL of deionized water. Absorbance was measured using UV-vis spectrophotometer (Spectronic Genesys^®^, with Winspec Software, Spectronic, (Pittsford, NY, USA) at 515 nm using water as blank. The amount of the free BB extract (un-loaded) in the supernatant was determined. The percentage of the entrapped *BBE* (*EE* %) was calculated from Equation (1) as follows:(1)$$EE\;\left(\%\right)=\frac{Amount\;of\;BBE\;added-Amount\;of\;BBE\;lost}{Amount\;of\;BBE\;added}\times 100\;$$

The experiment was carried out in triplicate and the average was calculated.

### 2.9. Hepatic Ischemia/Reperfusion Surgery Model

Male wistar rats weighing 200–240 g were obtained from the Animal Facility, Nahda University in Egypt. “The Commission on the Ethics of Scientific Research”, Faculty of Pharmacy, Minia University approved the research (License No. ES02/2021). As previously explained, hepatic ischemia was established [22]. Briefly, Wistar rats were anesthetized with (1 g/kg) i.p injection of urethane hydrochloride and a midline laparotomy was done to identify and clamp the portal vein, hepatic artery, and hepatic duct. After 30 min, the clip was withdrawn to begin hepatic reperfusion, which lasted 2 h.

### 2.10. Experimental Design

The study was performed on 70 male wistar rats randomly allocated to seven groups of ten rats each, (*n* = 10) (Figure 1):

Group I (Sham): sham-operated control group, animals subjected to the surgery procedures but without clamping.

Group II (IRI): hepatic IRI group.

Group III (*BBE*): animals treated with blackberry leaves extract (200 mg/kg) orally for 14 days then animals had hepatic IRI [26,45].

Group IV (AgNPs): animals treated with empty silver nanoparticles (100 mg/kg) orally for 14 days then animals had hepatic IRI.

Group V (200 *BBE*-AgNPs): animals treated with blackberry leaves extract nanoparticles (200 mg/kg) orally for 14 days then animals had hepatic IRI.

Group VI (50 *BBE*-AgNPs): animals treated with blackberry leaves extract nanoparticles (50 mg/kg) orally for 14 days then animals had hepatic IRI.

Group VII (Silymarin): Positive control, animals treated with Silymarin (100 mg/kg) orally for 14 days then animals had hepatic IRI [46,47,48].

When the experimental procedures were completed, rats was sacrificed, blood was collected by cardiac puncture for serum analysis and liver tissue was harvested; both were stored at −80 °C until used.

### 2.11. Preparation of Tissue Homogenate

Liver tissues were cut into small pieces and homogenized in 5 mL phosphate buffer containing [0.5 g of Na_2_HPO_4_ in addition to 0.7 g of NaH_2_PO_4_ per 500 mL of deionized water (pH 7.4) per gram tissue] followed by centrifugation at 4000 rpm for 10 min at 4 °C. Supernatant was separated to evaluate oxidant and antioxidant markers.

### 2.12. Biochemical Analysis

#### 2.12.1. Determination of Liver Function Parameters

The collected sera were used for the estimation of ALT and AST according to previous methods of Reitman and Frankel [49] using the corresponding Biodiagnostic Laboratory colorimetric assay kits (Giza, Egypt).

#### 2.12.2. Assessment of Oxidant and Antioxidant Markers

The tissue levels of reduced GSH, SOD, CAT and MDA were estimated using Biodiagnostic Laboratory colorimetric assay kits (Giza, Egypt). All assessments were performed according to the manufacturers’ protocols.

### 2.13. Histopathological Study

To assess the histological alterations, Liver tissue samples were fixed in 10% buffered formalin solution for 24 h before being dehydrated in escalating degrees of ethanol, cleaned in xylene, and fixed with paraffin. These samples were cut into 4 μm thick slices and stained with hematoxylin and eosin (H & E), after which the pathological alterations were observed under a microscope (scale bar 200 μm) and evaluated by a professional observer who was not informed of the identity of the studied specimens. Histological changes were scored in a blind fashion from 0, no injury; 1, mild injury (25%); 2, moderate injury (50%); 3, severe injury (75%); and 4, very severe injury (almost 100%) based on the degree of central vein congestion, sinusoidal congestion, parenchymal cells necrosis, cytoplasmic vacuolization, and inflammatory cells infiltration using modified Suzuki classification [50]. The overall histology score was calculated by adding the scores for all parameters. Each rat had three slides made for examination. All photomicrographs in this work were obtained with an Olympus (U.TV0.5XC-3) light microscope and digital camera.

### 2.14. Western Blotting

For evaluation of cleaved capsase-3, p-PI3K, p-mTOR and relative p-Akt/Akt expression in liver tissue, immunoblotting was conducted as described before [51]. In brief, liver tissue was homogenized in RIPA lysis buffer, which comprised 25 mM Tris-HCl, pH 7.6; 150 mM NaCl; 1% NP-40; 1% sodium deoxycholate; 0.1% SDS and 1% Protease/phosphatase Inhibitor cocktail. The supernatant was produced by centrifuging the homogenate for 20 min at 4 °C and then storing it at −80 °C. After boiling for 5 min at 95 °C, samples (30 μg/lane) were run through a 10% SDS-PAGE gel. The nitrocellulose membranes were blocked in 7.5% skimmed milk in TBS-T (0.05% Tween-20 Tris-buffered saline) for 2 h at room temperature before being incubated with primary antibodies diluted at 1:1000 against cleaved caspase-3, p-PI3K, p-Akt, Akt, and p-mTOR overnight at 4 °C. Following that, the membranes were washed and probed for 1 h at room temperature with secondary alkaline phosphatase-conjugated Mouse/Rabbit IgG antibody, followed by repeated washing. Protein bands were eventually observed using a colorimetric detection approach based on 5-bromo-4-chloro-3-indolyphosphate (BCIP)/nitro-blue tetrazolium (NBT). Image-J (NIH, Bethesda, MD, USA) software was used to examine the quantification of the identified bands. As a loading control, protein loading was adjusted for β-actin.

### 2.15. Immunohistochemistry

Immunohistochemical expression (IHC) was performed using polyclonal antibody for Bax and Cleaved Caspase-9. Liver tissue samples were fixed in 10% buffered formalin solution and embedded in paraffin. Deparaffinized and well hydrated sections (4 μm) were cooked for 10 min in a 25 mM citrate buffer solution (pH 6.0), then passed to boiling deionized water and allowed to cool for 20 min. Tissue slices were treated with 3% H_2_O_2_ to inhibit endogenous peroxidase activity. Slides were treated with 1% fetal bovine serum and 10% rat serum for 1 h at room temperature to inhibit non-specific immune-staining. The slides were treated overnight at 4 °C with anti-Bax and anti-Cleaved Caspase-9 antibodies. Subsequently, the slices were then stained with a solution of DAB (3-3-diaminobenzidine) after being treated for 30 min at 37 °C with biotinylated goat anti-rabbit IgG secondary antibody (Invitrogen). After 10 s of hematoxylin staining, the slides were mounted. Negative controls were used to assess the technique’s specificity (canceling the incubation with the primary antibody and incubating it with non-immune sera). To measure the positive areas, the programme Image J (NIH, Bethesda, MD, USA) was employed. Results were expressed as the percentage of stained cells in each field [52].

### 2.16. In Silico Study

#### 2.16.1. Ischemia Potential Protein Targets Determination

As a way to propose protein targets for the *BBE*-identified compounds, we ran inverse docking simulations on each chemical against every protein in the Protein Data Bank. In order to accomplish this, we made use of the idTarget platform (http://idtarget.rcas.sinica.edu.tw/; accessed on 15 September 2022). This structure-based screening platform employs a novel docking strategy, called divide-and-conquer docking, which adaptively constructs small overlapping grids to limit the search space on protein surfaces. As a result, it can conduct a large number of high-quality docking experiments in a significantly shorter amount of time [53]. The data were compiled as a table of binding affinity scores, which were then ranked from the highest negative value to the lowest one.

Using a cutoff of −10.0 kcal·mol kcal/mol for binding affinity, we determined which targets were optimal for each molecule annotated in *BBE*. A total of 11 protein targets were identified for compounds 1–13 of which PLA_2_ (PDB: 5WZS) was found to be the most relevant target to ischemia-reperfusion. 

#### 2.16.2. Molecular Dynamic Simulation and Binding Free Energy Calculation

Previous report for calculating the binding free energy (Δ*G*) and performing molecular dynamic simulations were followed [54]. These procedures are described in detail in the Appendix A.

### 2.17. Statistical Analysis

All numerical results were presented as mean ± SEM for 6 rats in each group and analyzed with GraphPad prism version 9 (San Diego, CA, USA). To analyze differences across all experimental groups, One-way ANOVA test was employed, followed by a Tukey-Kramer post hoc test. The significance level was set at *p* < 0.05.

## 3. Results

### 3.1. LC–HR–ESI–MS Chemical Profiling of BBE

Metabolomic profiling of *BBE* using LC-HR-MS for dereplication purposes has led to the identification of a number of natural products (Table 1 and Figure 2), of which Flavonoids were identified to predominate (1–6) followed by phenolic acids (9–12). LC-HRMS analysis was performed using both positive and negative ionization modes to cover a wide range of compounds in *BBE* (Appendix A). In addition, two anthocyanins (6 and 7) and one alkaloids (13) were also identified as major compounds in *BBE*. All of these annotated natural products have been previously reported from black berry [55].

### 3.2. Characterization of the Prepared BBE-AgNPs Nanoparticles

Silver nitrate nanoparticles of blackberry extract (*BBE*-AgNPs) were successfully prepared. That was obvious due to the yellow then reddish discoloration of the colorless pectin on reducing silver ions. Moreover, the given UV peak at 410 nm due to surface plasmon resonance confirmed the formation of AgNPs. (Figure 3) shows the spherical Nano-structure of the formed nanoparticles. Formulated nanoparticles were homogenously distributed within Nano-sized range (37.2 nm to 300.8 nm). The homogenous distribution could be attributed to the reducing power of PVP allowing the formation of tiny clusters of AgNPs. This obvious variation in particle size of different formulations could be attributed to the different formulation parameters. Increasing the amount of pectin, relatively to AgNO3, in F2 compared to F1 has resulted in reduced particle size from 190.3 ± 5.6 to 37.2 ± 1.3, respectively (Table 2). This could be due to the dispersing effect of pectin that prevents the agglomeration of AgNPs. On the other hand, increasing the amount of pectin above certain limit may hinder the formation of small nanoparticles as reported by Natsuki et al. [56].

### 3.3. Entrapment Efficiency of the Prepared BBE-AgNPs Nanoparticles

The amount of the free *BBE* in the supernatant was determined. The percentage of the entrapped *BBE* (*EE* %) was calculated from (Equation (1)) as showed in Table 2.

F2 was selected for biological study due to its small particle size which allow enhanced cellular uptake to hepatic cells. Zeta potential of the selected formula was—32.5 mv which indicates the improved physical stability of the prepared AgNPs and that the loaded *BBE* did not affect the charge or the stability of the prepared nanoparticles.

### 3.4. Serum ALT and AST Activities

It was revealed that the hepatic I/R-induced injury is associated with a substantial elevation in serum ALT (3.9 fold) (Figure 4A) and AST (4.7 fold) (Figure 4B), as compared to the sham rats. However, treatment with 200 *BBE* significantly leveled off these markers. Additionally, pretreatment with 200 *BBE* loaded AgNPs and 50 *BBE* loaded AgNPs has prevented the I/R effect and returned these enzymes at their normal level with the lower dose showing a better effect to match that of Silymarin. This drug targeting effect of nanoparticles superimposed that *BBE* treatment alone. The results are presented in Appendix A.

### 3.5. Hepatic Lipid Peroxidation and Antioxidant Markers

As shown in Figure 5, the hepatic I/R-induced injury is associated with a significant decrease in antioxidant parameters (Figure 5A) GSH (80%), (Figure 5B) SOD (50%), (Figure 5C) CAT (63%) and increase in (Figure 5D) MDA (1.8 fold) which is a marker for lipid peroxidation, oxidative stress, and tissue injury, as compared to that of the sham rats. However, treatment with 200 *BBE* significantly normalized these markers. Additionally, pretreatment with 200 *BBE*-AgNPs and 50 *BBE*-AgNPs has inhibited the I/R effect and kept these parameters at their normal level, with the lower dose showing a better effect to match that of Silymarin. The results are presented in Appendix A.

### 3.6. Histopathological Analysis

The microscopic examination (Figure 6) of (A) sham rats showed normal hepatic architecture (non-dilated and non-congested central veins (arrow) surrounded by hepatocytes arranged in normal cords and separated by hepatic sinusoids without vacuolation (arrow head) (grade 0). Nonetheless, sections of the hepatic IRI group reveals (B) focal pale eosinophilic area with small dark pyknotic nuclei represent 25% of liver tissue (arrow), peripheral dilated and congested central vein surrounded by dilated and congested sinusoids (arrow head), (C) marked dilated and congested central vein and sinusoids (arrow), The Surrounding Hepatocytes are hexagonal shaped with mild vacuolation arrow head, (D) Marked dilated and congested portal vein arrow surrounded by mixed acute and chronic inflammatory cellular infiltrates formed mainly of lymphocytes and plasma cells arrow head, moderate cytoplasmic vacuolization of hepatocytes (strikes). (E) Slight improvement in *BBE* group showing moderate dilated and congested central veins (arrow) surrounded by hepatocytes arranged in cords and separated by hepatic sinusoids without vacuolation with small focal area of hepatic necrosis 5% (arrow head). (F) AgNPs group exhibiting moderate dilated and congested central veins and sinusoids (arrow). The Surrounding Hepatocytes are hexagonal shaped with moderated vacuolation of fatty change (arrow head). (G) Significant improvement was observed in 200 *BBE*-AgNPs group micrograph that have moderate dilated and congested central veins (arrow) surrounded by hepatocytes arranged in cords and separated by hepatic sinusoids without vacuolation (arrow head). (H) much better improvement was observed in 50 *BBE*-AgNPs group micrograph that show mild dilated and congested central veins (arrow) surrounded by hepatocytes arranged in cords and separated by mild dilated and congested hepatic sinusoids without vacuolation (arrow head). (I) Silymarin group shows marked regression of the histopathological lesions with mild dilated and congested central veins (arrow).

A better effect is detected in sections of 50 *BBE* AgNPs and 200 *BBE* AgNPs treated groups with the lower dose showing a better effect comparable to silymarin which reveals mild histopathological alterations (Figure 6 and Table 3).

### 3.7. Effect of BBE-AgNPs on Cleaved Caspase-3, p-PI3k, p-Akt, p-mTOR Protein Expression

As depicted in western blotting, induction of hepatic I/R sharply reduced the hepatic protein expression of phosphorylated PI3K, Akt and mTOR, by 85.8%, 74.5% and 81.2%, respectively, compared to those of sham group, as well as elevated the protein expression of casp-3 by (4.2 fold). Pre-administration of *BBE* promoted the phosphorylation PI3K, Akt and mTOR, an effect that was further augmented to reach almost 4.7, 2.1 and 3.1 folds respectively by 200 *BBE*-AgNPs and 6.1, 3.5 and 4.3 folds respectively by 50 *BBE*-AgNPs, compared to the IRI group. *BBE* caused a subtle yet significant decrease in the protein expression of Casp-3, while both doses of *BBE*-AgNPs revealed a drug targeting effect showing better effects, compared to the IRI, *BBE* and empty AgNPs pretreated groups. Collectively, results explain that the nano formulation enhanced the drug targeting effect of *BBE* and maintained the protein expressions at the normal level (Figure 7 and Figure 8). Original images of western blotting are attached in the Appendix A.

### 3.8. Effect of BBE-AgNPs on Bax and Cleaved Caspase-9 Protein Expression

As observed in (Figure 9), the immunohistochemical imaging revealed that the Sham group shows negative hepatic Bax and caspase-9 immunoreactivity contrary to those of the I/R and AgNPs groups that have strong intensity of cytoplasmic staining revealing strong immune expression (8.5 and 7.8 folds) respectively for Bax and (12.4 and 11 folds) and for cleaved caspase-9 compared to Sham group. Nonetheless, a moderate Bax and cleaved caspase-9 expression is seen in section of *BBE* treated rats, whereas weak expressions are recorded on 200 or 50 *BBE*-AgNPs pretreated groups that simultaneously decreased Bax and caspase-9 expressions by 64%, 60% for 200 *BBE*-AgNPs treated group and by 75%, 66.6% for 50 *BBE*-AgNPs treated group. The results are presented in Appendix A.

### 3.9. In Silico Study

#### Identifying the Likely Molecular Target of *BBE*’s Dereplicated Natural Products

The modelled structures of the dereplicated compounds (Table 1, Figure 2) in *BBE* was put through a docking-based virtual screening to identify the likely molecular target(s) via which it can mediate its liver protective effect. You can access the idTarget online system at https://idtarget.rcas.sinica.edu.tw (accessed on 14 February 2023) [53]. The vast majority of the proteins available through the Protein Data Bank (PDB; https://www.rcsb.org/ (accessed on 22 November 2021) can be screened virtually using this platform’s inverse docking method. An Excel file was generated from the obtained results, with the binding affinity scores ordered from highest negative value to lowest. The optimal result was chosen using a cut-off value of −10.0 kcal/mol.

Human Phospholipase A2 (PLA2) (PDB code: 5WZS) [57] was found to be the most relevant target associated to the liver dysfunction upon induction of ischemia-reperfusion [58,59,60]. Both cyanidin and pelargonidin glucosides (the major anthocyanins of *BBE*; Table 1 and Figure 2) were the compounds that showed the best docking scores with this target protein (−12.67 and −12.58 kcal/mol). PLA2 over expression occurred in liver cells as a result of ischemia-reperfusion indirectly induce the activity of the oxidant enzymes COX and LOX that in turn, increase the intracellular oxidative stress. From another side PLA2 induce cardiolipin hydrolysis that in turn, leads to mitochondrial dysfunction and also increased intracellular oxidative stress (Figure 10). Besides their powerful intrinsic antioxidant effect, black berry’s anthocyanins have been shown to be effective PLA2 inhibitors [61,62,63,64].

To further validate the preliminary docking findings, the generated docking poses of both cyanidin and pelargonidin glucosides inside the PLA2’s active site were subjected to 50 ns-long MDS experiments to study their dynamic modes of interaction and to estimate their binding affinities (i.e., absolute binding free energy, Δ*G*_binding_). As shown in (Figure 11), Both cyanidin and pelargonidin glucosides were able to achieve stable bindings inside the human PLA2’s active site via forming three stable H-bonds with ASN-1, ASP-47, and GLY-30 similarly to the co-crystalized inhibitor. In addition, they established π-staking interactions with PHE-5 together with co-ordinate interactions with Ca^2+^ through their phenolic hydroxyl groups. As a result, their deviations (i.e., RMSD) from the initial docking poses were only about ~2.52 Å (RMSD of the co-crystalized inhibitor ~2.76 Å), and their calculated absolute binding free energy (Δ*G*_binding_) were −9.56 and −9.58 kcal/mol, respectively (Δ*G*_binding_ of the co-crystalized inhibitor = −8.78 kcal/mol) indicating significant affinity to the enzyme’s active site. Previously, a number of anthocyanins and anthocyanidines were found to act as potent PLA2 inhibitors both in vitro and in vivo [61,62].

Accordingly, the previous in silico-based investigation suggested that *BBE*-derived anthocyanins (i.e., cyanidin and pelargonidin glucosides) are a promising scaffold for developing potent anti-inflammatory and antioxidant agents targeting PLA2. However, careful in vitro and in vivo testing of the plant’s isolated anthocyanins for their PLA2’s inhibitory activity should be carried out in future biological investigations of this plant.

## 4. Discussion

Hepatic IRI, in addition to hemorrhagic shock, is a condition in which cellular damage occurs frequently during liver transplantation, partial hepatic resection, and trauma circumstances [65]. The current emphasis on herbal supplements motivates scientists to research natural agents in a variety of illnesses. Several researches have demonstrated that natural drugs can play an important role in human diseases prevention. Among these is blackberry extract, which has been shown to be anti-carcinogenic, anti-inflammatory, antimicrobial anti-diabetic, and antiviral [66,67]. Blackberry leaf extract is a natural therapeutic medication with many regulatory actions, including improved microcirculation, suppression of apoptosis, and inhibition of neutrophil infiltration and inflammation [68]. For many years, blackberry-based therapy has been utilized for hepatic disorders and as a potent antioxidant [45,69]. Unfortunately, it is only effective at high doses which limited its clinical utility [45].

The optimization of a nano-sized drug delivery system has the ability to significantly improve the therapeutic efficacy of the loaded drug [70,71,72,73,74]. The goal of this work was to minimize particle size and improve the *EE%* and medication targeting action of *BBE*-AgNPs nano-formulation. To improve therapeutic efficacy, blackberry-based therapy is implemented by incorporating the extract into AgNPs nano-architecture to accomplish targeted therapy. It should be mentioned that, depending on the AgNPs quantities and schedule of administration, silver nanoparticles can have both protective and harmful effects [75,76,77]. Therefore, the therapeutic window for I/R injury in the liver should be thoroughly investigated. It should be taken into consideration that reported studies have already found that liver accumulated the highest level of AgNPs in animals which make it an ideal nano-carrier to the liver [78,79].

In this study, we developed a *BBE* Loaded AgNPs (*BBE*-AgNPs) formulation that contained the antioxidant blackberry leaves extract constituents to protect the liver from I/R injury. Besides, silymarin, a natural hepato-protective agent, was used as a positive control because it was reported that silymarin protects liver tissue in hepatic I/R injury in rats by regulating the apoptotic pathway and modulating the antioxidant status [80]. Current results demonstrated that *BBE* could significantly decrease the elevated level of serum hepatic enzymes (ALT&AST) in IRI-induced in rats. Additionally, *BBE*-AgNPs showed restoration of hepatic function similar to that of silymarin, and with lower dose. In hepatic cells, when the cellular membrane of the liver cells was damaged, the permeability of the membrane would be increased, resulting in the leakage of the liver enzyme (ALT) in hepatic cytoplasm into circulation and a rapid raise of serum ALT level [20]. The unremitting hepatic injury also leads to a discharge of AST in hepatic cytoplasm and the mitochondria into the circulation [81]. Parallel with the elevated ALT and AST serum levels, Suzuki’s score were increased in hepatic IRI as showed severe hepatic necrosis, vacuolar degeneration and inflammatory cellular infiltrates that accompanied by congested and dilated hepatic central vein and sinusoids. All aforementioned damages were reduced by *BBE*-AgNPs pretreatment with a posterior effect for the lower dose. These results suggest that *BBE* loaded AgNPs may exert hepatoprotective effect after I/R injury.

The process of I/R injury disrupts redox balance and damages normal tissue functions and structure, culminating in the accumulation of ROS. As a result, the most common category liver ischemia-reperfusion injury mechanism is the formation of ROS and oxidative stress [82]. The role of oxidative stress in ischemia and reperfusion injury is widely acknowledged. Here, IRI group exhibited a significant decrease of antioxidants (SOD, CAT, and GSH) with elevation of MDA as a lipid peroxidation marker. SOD has the ability to increase the conversion of O_2_^−^ to O_2_ and H_2_O_2_. Under CAT catalysis, H_2_O_2_ can also be degraded into H_2_O and O_2_. As a result, both enzymes were found to be helpful by hastening the detox of ROS in IRI [69]. The initial line of defense against oxidative stress is antioxidants such as SOD, CAT, and GSH [83]. This study found that *BBE*-AgNPs mediated the aforementioned antioxidant markers linked with IRI and had a mitigative effect against oxidative stress-induced damage which is observed by decreased MDA level. The *BBE* components SOD/CAT-like activities may scavenge O_2_^−^ and H_2_O_2_ to create O_2_ by employing the excessive ROS generated during hepatic ischemia/reperfusion as a trigger. Such a cascade reaction efficiently performed ROS scavenging (O_2_^−^ and H_2_O_2_) and reducing oxidative stress during hepatic ischemia and reperfusion [84]. Numerous cellular pathways, including the protective (PI3K/Akt/mTOR) pathway, are suppressed during hepatic I/R shock, while several harmful processes, such as oxidative stress and apoptosis, are activated [22].

As illustrated in (Figure 12), The Phosphatidylinositol-3-kinase (PI3K) pathway, which includes Akt, mTOR, is activated by the dimerization of tyrosine kinase receptor that activated by growth factors. It represents an important component in regulating apoptosis and pro-inflammatory cytokines [85]. Hepatic I/R damage is linked to a number of signaling pathways. Recent study found that activated phosphorylated Akt (p-Akt) dramatically mitigate I/R injury to the liver. Apoptosis, also known as type I programmed cell death, is linked to hepatic I/R injury. PI3K/Akt signaling activation protects hepatic cells from apoptosis [86].

The current findings revealed that *BBE*-AgNPs in multiple doses upregulated the phosphorylation of PI3K, Akt, and mTOR proteins of the protective signaling pathway, and reduced the ischemic injury more effectively than *BBE* alone. Previous studies showed that the overproduction of ROS and inhibition of PI3K/Akt/mTOR pathway led to the up-regulation of Bax, Casp-9 and Casp-3 expressions, resulting in apoptosis [86,87]. Bcl-2 associated X (Bax) proteins are proapoptotic molecules that induce mitochondrial cytochrome C release, forming apoptotic protease activating factor 1 (APAF-1) and hence boosting caspase-9 and caspase-3 production [88]. Caspase-3 is one of the main molecules involved in signaling apoptosis. When the apoptotic cascade is triggered, procaspases are cleaved to release the active caspases. Therefore, cleaved caspase-3 is considered a reliable biomarker for apoptosis [89]. The expression of Bax, cleaved casp-9 and casp-3 proteins was elevated in the IRI rats. Bax alone has been found to be adequate for apoptotic induction [90]. Apoptosis was observed to be related with caspase-3 activation and Bax expression during graft cold storage and after cold I/R in rat liver transplantation [91]. Apoptosis of hepatocytes and sinusoidal endothelial cells is an important cause of liver I/R injury [92].

Interestingly, Bax, casp-9 and casp-3 proteins were down-regulated after the treatment with *BBE*-AgNPs in both doses more than BEE alone. Taken together, we demonstrated that *BBE*-AgNPs in both doses mitigated the hepatic ischemic damage and limited apoptosis by suppressing Bax, casp-9 and casp-3 proteins expression.

Furthermore, it can be deduced that using 50 *BBE*-AgNPs yielded better outcomes for all treatment groups of experimental rats than the 200 *BBE*-AgNPs. This may be due to the hepatotoxic effect of relatively higher doses of AgNPs as explained by previous studies [75,93,94]. Vasquez et al. measured the effects of AgNPs on hepatic enzymes after a single dose of Paracetamol (2 g/kg BW) and revealed that the smaller dose of AgNPs (100 mg/kg BW) are less toxic than larger dose of AgNPs (200 mg/kg BW) [75]. Also Hamad et al. observed that sections of the livers of larger dose AgNPs (200, 300 mg/kg) treated mice showed vascular congestion and scattered inflammatory cells (lymphocytes) which indicate the initiation of the inflammation. On contrast smaller dose AgNPs (50, 100 mg/kg) administration showed no histopathological changes [93].

## 5. Conclusions

Finally, we conclude that by using nanotechnology, *BBE* herbal extract can be loaded to nanoparticles, that ultimately enhances absorbability together with bioavailability toward hepatic IRI and enhances therapeutic outcome with a relatively low dosage, which is likely attributed to improved pharmacokinetics and tissue distribution [95]. Our findings suggest that *BBE*-AgNPs can protect against mitochondrial apoptosis and oxidative stress caused by liver I/R, which make *BBE*-AgNPs a potential therapeutic candidate. In-depth, further understanding of the molecular pathways of IRI is critical for improving ongoing therapeutic regimens and developing novel appropriate interventions. Furthermore, attempts should be made to load other natural extracts into nano-carriers, which may enhance effectiveness and patient survival rate. Collectively, this study offers a viable approach of multifunctional nano-therapeutics for IRI treatment, and it also highlights the dual protection mechanism of *BBE* via ROS clearance and apoptosis regulation.

## Figures and Tables

**Figure 1 metabolites-13-00419-f001:**
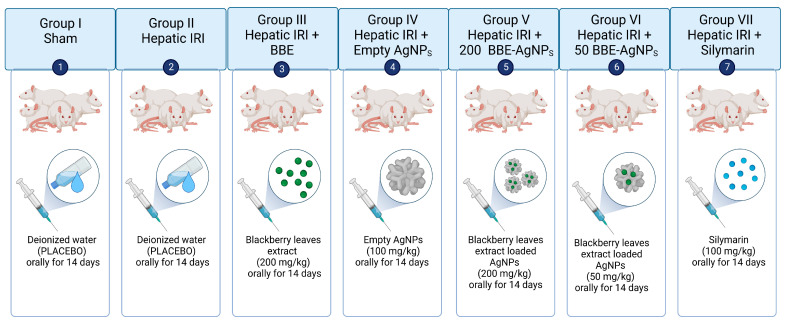
The Experimental Design (created with BioRender.com).

**Figure 2 metabolites-13-00419-f002:**
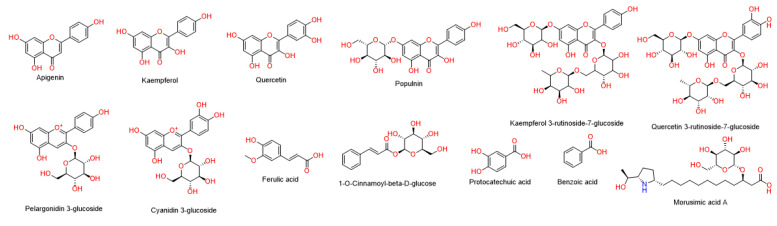
Annotated compounds in *BBE*.

**Figure 3 metabolites-13-00419-f003:**
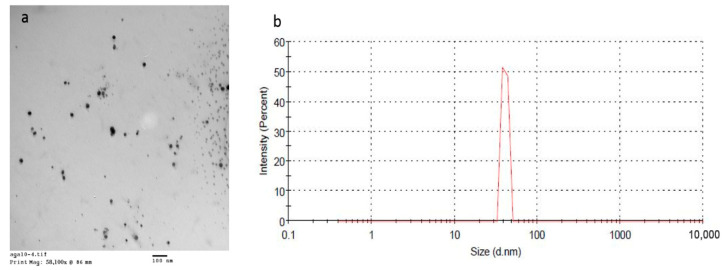
(**a**) TEM and, (**b**) size distribution of *BBE*-AgNPs (F2).

**Figure 4 metabolites-13-00419-f004:**
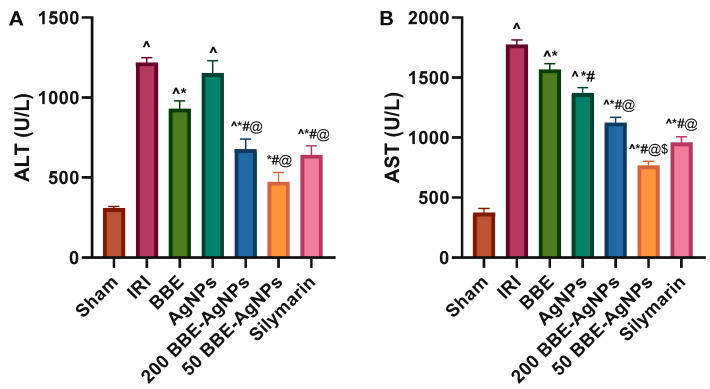
Effect of pretreatment with *BBE*, AgNPs, 200 or 50 *BBE*-AgNPs and silymarin on serum (**A**) ALT and (**B**) AST level in hepatic I/R injured rats. Values are presented as means of 6 animals ± SEM. Data were analyzed using one-way ANOVA followed by Tukey’s Multiple Comparisons Test; *p* < 0.05. As compared with (^) Sham, (*) IRI, (#) *BBE*, (@) AgNPs and ($) 200 *BBE*-AgNPs. ALT, alanine transaminase; AST, aspartate transaminase; *BBE*, Blackberry Extract; IRI, ischemia/reperfusion injury; AgNPs, Silver Nanoparticles; *BBE*-AgNPs, Blackberry loaded Silver Nanoparticles.

**Figure 5 metabolites-13-00419-f005:**
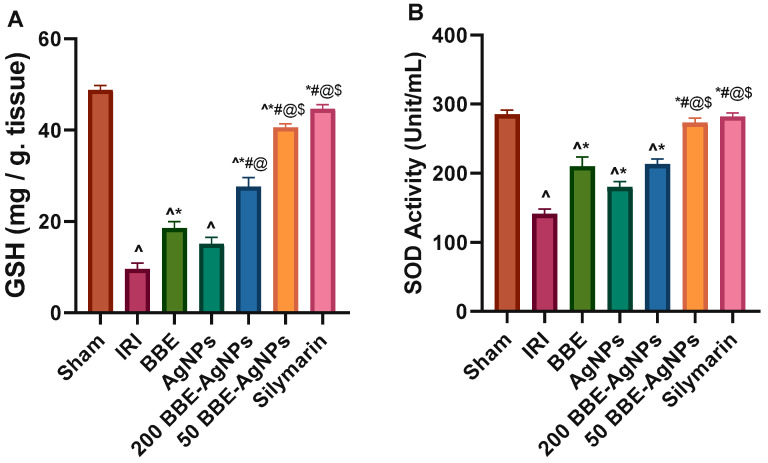
Effect of pretreatment with *BBE*, AgNPs, 200 or 50 *BBE*-AgNPs and Silymarin on hepatic content of (**A**) GSH, (**B**) SOD, (**C**) CAT and (**D**) MDA level in hepatic I/R injured rats. Values are presented as means of 6 animals ± SEM. Data were analyzed using one-way ANOVA followed by Tukey’s Multiple Comparisons Test; *p* < 0.05. As compared with (^) Sham, (*) IRI, (#) *BBE*, (@) AgNPs and ($) 200 *BBE*-AgNPs. ANOVA, analysis of variance; GSH, reduced glutathione; CAT, catalase; SOD, superoxide dismutase; MDA, malondialdehyde; *BBE*, Blackberry Extract; IRI, ischemia/reperfusion injury; AgNPs, Silver Nanoparticles; *BBE*-AgNPs, Blackberry loaded Silver Nanoparticles.

**Figure 6 metabolites-13-00419-f006:**
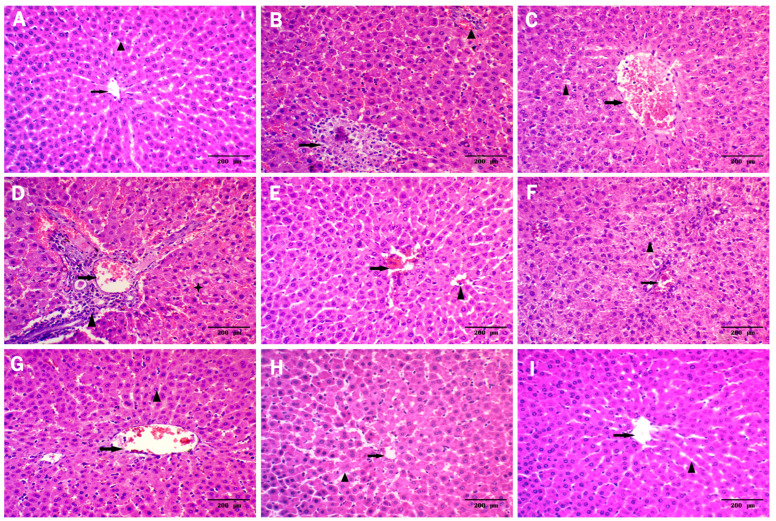
Effect of pretreatment with *BBE*, AgNPs, 200 or 50 *BBE*-AgNPs and silymarin on histologic changes induced by hepatic I/R injury in rats. Photomicrographs represent (**A**) Sham, (**B**–**D**) IRI, (**E**) *BBE*, (**F**) AgNPs, (**G**) 200 BEE-AgNPs, (**H**) 50 *BBE*-AgNPs and (**I**) Silymarin.

**Figure 7 metabolites-13-00419-f007:**
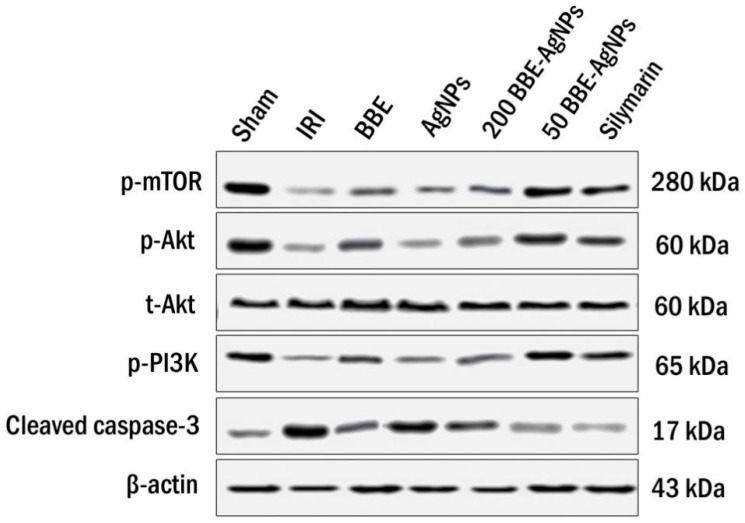
Representative western blot analysis. Bands of the measured proteins in liver of the studied groups. Bands represent Sham, IRI, *BBE*, AgNPs, 200 BEE-AgNPs, 50 *BBE*-AgNPs and Silymarin.

**Figure 8 metabolites-13-00419-f008:**
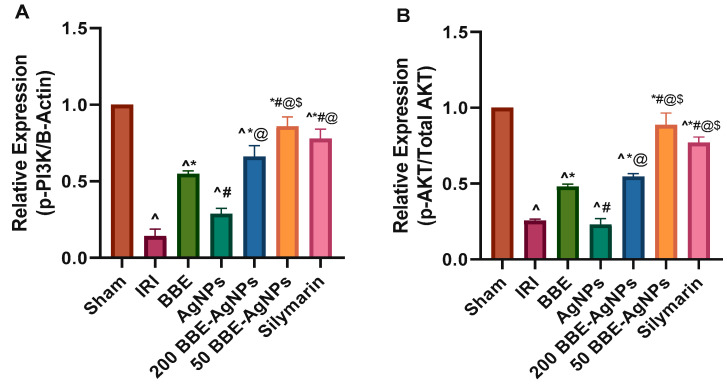
Quantitative analysis of affected proteins showing the effect of pretreatment with *BBE*, AgNPs, 200 or 50 *BBE*-AgNPs and Silymarin on the relative protein expression of (**A**) p-PI3K, (**B**) p-AKT, (**C**) p-mTOR and (**D**) cleaved caspase-3 in I/R injured rats. Values are presented as means of 3 independent results ± SEM. Data were analyzed using one-way ANOVA followed by Tukey’s Multiple Comparisons Test; *p* < 0.05. As compared with (^) Sham, (*) IRI, (#) *BBE*, (@) AgNPs and ($) 200 *BBE*-AgNPs groups. PI3K, Phosphatidylinositol-3-kinase; p-Akt, protein kinase B phosphorylated at serine 473; p-mTOR; phosphorylated mammalian target of rapamycin; ANOVA, analysis of variance; *BBE*, Blackberry Extract; IRI, ischemia/reperfusion injury; AgNPs, silver nanoparticles; *BBE*-AgNPs, Blackberry loaded silver nanoparticles.

**Figure 9 metabolites-13-00419-f009:**
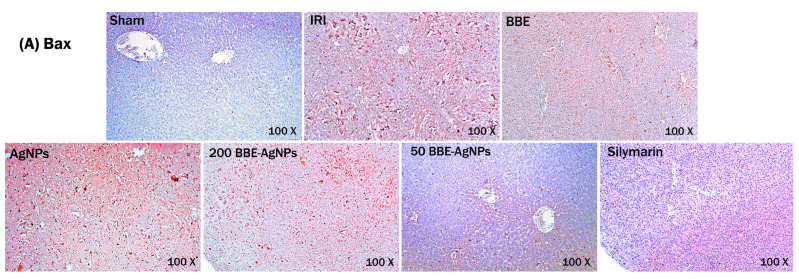
Effect of pretreatment with *BBE*, AgNPs, 200 or 50 *BBE*-AgNPs and Silymarin on the on apoptosis of hepatic cells in I/R injured rats. Representative photomicrographs of (**A**) Bax and (**B**) Casp-9 proteins immunohistochemical staining indicates expression of Bax and Cleaved Caspase-9 in liver tissue. Positive staining was brown. (magnification = 200×); (**C**) % area of positive Bax staining; (**D**) % area of positive cleaved caspase-9 staining Photomicrographs represent Sham, IRI, *BBE*, AgNPs, 200 BEE-AgNPs, 50 *BBE*-AgNPs and Silymarin. values are expressed as means of 6 animals ± SEM. Data were analyzed using one-way ANOVA followed by Tukey’s Multiple Comparisons Test; *p* < 0.05. As compared with (^) Sham, (*) IRI, (#) *BBE*, (@) AgNPs and ($) 200 *BBE*-AgNPs groups. Bax, bcl-2-associated x protein; Casp-9, Cleaved Caspase-9; ANOVA, analysis of variance; *BBE*, Blackberry Extract; IRI, ischemia/reperfusion injury; AgNPs, silver nanoparticles; *BBE*-AgNPs, Blackberry loaded silver nanoparticles.

**Figure 10 metabolites-13-00419-f010:**
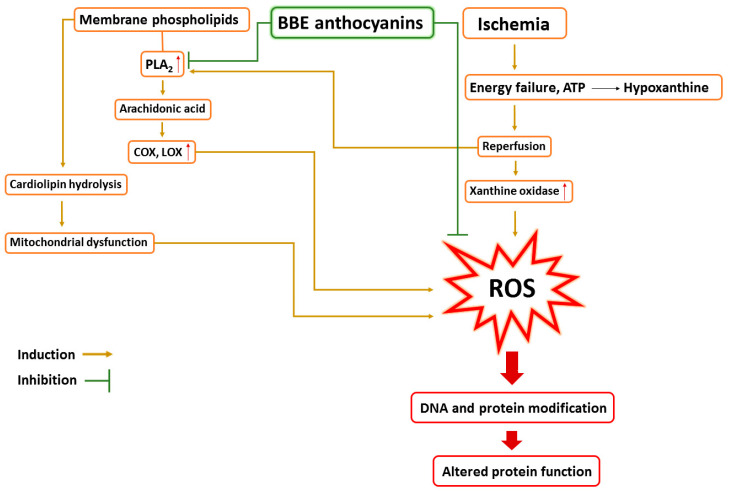
Schematic representation of the role of ischemia-reperfusion in the induction of ROS in liver cells showing the role of PLA2 and the possible protective effect of *BBE*’s anthocyanins. PLA2, Phospholipase A2; COX, Cyclooxygenase; LOX, Lipoxygenase; ATP, adenosine triphosphate; ROS, reactive oxygen species.

**Figure 11 metabolites-13-00419-f011:**
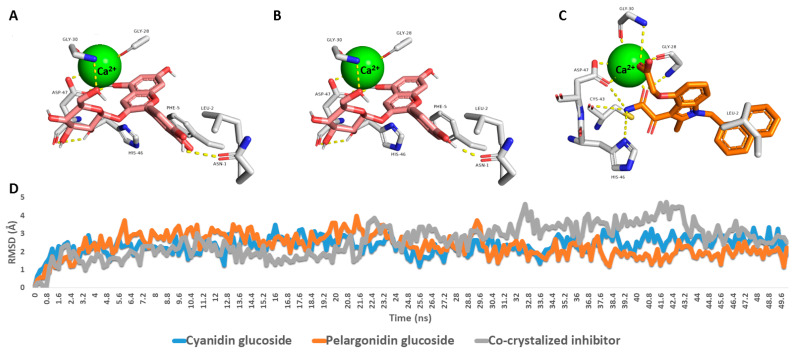
(**A**–**C**) Binding modes of the MDS-derived most populated poses of both cyanidin and pelargonidin glucosides along with the co-crystalized inhibitor (i.e., 7w9) inside human PLA2’s active site (PDB ID: 5WZS). (**D**) RMSDs of both cyanidin and pelargonidin glucosides along with the co-crystalized inhibitor over the course of 50 ns-long MDS runs.

**Figure 12 metabolites-13-00419-f012:**
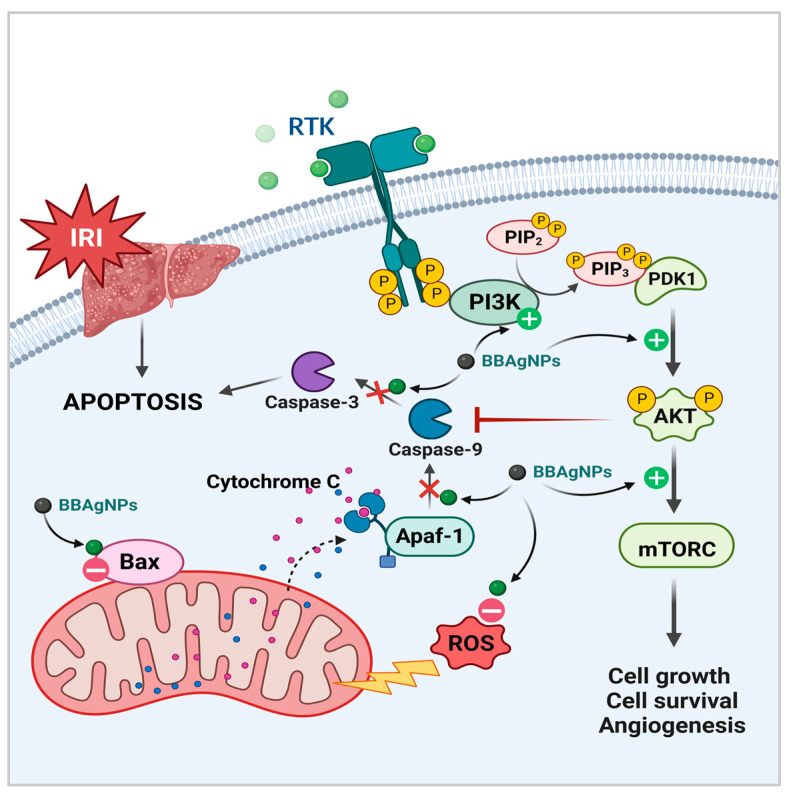
A hypothetical scheme of *BBE*-loaded AgNPs protection to hepatic ischemia/reperfusion injury (IRI). When the liver is subjected to ischemia/reperfusion, the expression of Bax is strongly induced, which leads to release of cytochrome c from mitochondria and formation of Apaf-1 which activate Caspase-9 and then Caspase-3 in hepatocytes leading to apoptosis. On the other hand, the inhibited PI3K/Akt/mTOR pathway inhibits cell growth and cell survival. Therefore, the PI3K/Akt/mTOR pathway activation by *BBE*-AgNPs components may regulate the expression of genes closely associated with apoptosis such as Bax, Caspase-9 and Caspase-3 leading to attenuation of liver injury after IR (created with BioRender.com).

**Table 1 metabolites-13-00419-t001:** LC-HRMS-Identified compounds in *BBE*. M.F., molecular formula.

No.	Identified Compound	Ionization Mode	*m*/*z*	M.F.	Accurate Mass	Calculated Mass	Chemical Class
1	Apigenin	Positive	271.0606	C_15_H_10_O_5_	270.0529	270.0528	Flavonoids
2	Kaempferol	Positive	287.0557	C_15_H_10_O_6_	286.0479	286.0477	Flavonoids
3	Quercetin	Positive	303.0508	C_15_H_10_O_7_	302.0430	302.0427	Flavonoids
4	Populnin	Positive	449.103	C_21_H_20_O_11_	448.1001	448.1006	Flavonoids
5	Kaempferol 3-rutinoside-7-glucoside	Positive	757.2194	C_33_H_40_O_20_	756.2116	756.2113	Flavonoids
6	Quercetin 3-rutinoside-7-glucoside	Positive	773.2138	C_33_H_40_O_20_	772.2060	772.2062	Flavonoids
7	Pelargonidin 3-glucoside	Positive	433.1126	C_21_H_21_O_10_	433.1126	433.1129	Anthocyanin
8	Cyanidin 3-glucoside	Positive	449.1080	C_21_H_21_O_11_	449.1080	449.1078	Anthocyanin
9	Ferulic acid	Negative	193.0496	C_10_H_10_O_4_	194.0574	194.0579	Phenolic acids
10	1-O-Cinnamoyl-beta-D-glucose	Negative	309.0971	C_15_H_18_O_7_	310.1050	310.1053	Phenolic acids
11	Protocatechuic acid	Negative	155.0347	C_7_H_6_O_4_	154.0269	154.0266	Phenolic acids
12	Benzoic acid	Negative	123.0444	C_7_H_6_O_2_	122.0366	122.0368	Phenolic acids
13	Morusimic acid A	Negative	490.3015	C_24_H_45_NO_9_	491.3095	491.3094	Alkaloid

**Table 2 metabolites-13-00419-t002:** Formulation parameters and characterization of *BBE*-AgNPs. Values of particle size, PDI and *EE%* are presented as means of 3 samples ± SEM. AgNO_3_, silver nitrate; PDI, Poly dispersity index; *EE%*, Entrapment Efficiency.

Formula Number	AgNO_3_(g)	Pectin (g)	Particle Size (nm)	PDI	*EE%*	Z-Potential (mv)
1	0.104	0.1	190.3 ± 5.6	0.35 ± 0.01	41.8 ± 3.2	−25.6
2	0.2	0.4	37.2 ± 1.3	0.37 ± 0.02	39.3 ± 2.5	−32.5
3	0.05	0.209	248.9 ± 12.5	0.59 ± 0.005	37.5 ± 5.4	−27.6
4	0.0725	0.4	300.8 ± 6.5	0.40 ± 0.007	38.1 ± 1.2	−30.4
5	0.2	0.145	220.2 ± 10.6	0.36 ± 0.01	40.6 ± 2.9	−25.3

**Table 3 metabolites-13-00419-t003:** Histopathological damage scores for liver tissues following hepatic ischemia reperfusion injury (IRI) of rats after the administration of *BBE*, AgNPs, 50 *BBE*-AgNPs, 200 *BBE*-AgNPs and Silymarin. IRI, ischemia/reperfusion injury; *BBE*, Blackberry Extract; AgNPs, silver nanoparticles; *BBE*-AgNPs, Blackberry loaded silver nanoparticles.

Degenerative Changes	Dilated and Congested Central Vein	Dilated Sinusoids	Hepatocytes Necrosis	Vacuolar Degenerations	Inflammatory Cellular Infiltrates
Sham	0	0	0	0	0
IRI	3	3	3	3	3
*BBE*	2	1	1	2	2
AgNPs	2	2	2	2	2
200 *BBE*-AgNPs	2	1	0	1	1
50 *BBE*-AgNPs	1	1	0	1	0
Silymarin	0	0	0	0	0

(0 = no change, 1 = mild change, 2 = moderate change, and 3 = severe change).

## Data Availability

Data generated or analyzed during this study are provided in full within the published article and its Appendix A.

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
