# Peer review of "Blackberry-Loaded AgNPs Attenuate Hepatic Ischemia/Reperfusion Injury via PI3K/Akt/mTOR Pathway"

_metabolites, 2023, doi:10.3390/metabo13030419_

Round 1

Reviewer 1 Report

This manuscript discusses IRI treatment by combining BBE with AgNPs and concludes: Through reducing PLA2 oxidative stress and apoptosis and activating the PI3K/Akt/mTOR signaling pathway, BBE may improve hepatic IRI that has been enhanced by BBE-AgNPs nano-formulation. The manuscript demonstrates that nanotechnology can load natural herbal extracts into nanoparticles, ultimately improving the absorption and bioavailability to target organs in vivo. However, the hypotheses deduced using some data should be verified by further experiments to be more in line with the rigor and reference value of the inference. The manuscript is attractive with academic value research and suitable for publication after modification.

Major comments:

1. The conclusion of the "Abstract" (Line 43) does not match the conclusion of the "Conclusion", and the meaning of the research results is not precise, and there is a gap.

2. Although the manuscript already cites research literature to support that anthocyanins can act as potent PLA2 inhibitors and deduces that BBE-derived anthocyanins (i.e., anthocyanins and pelargonidin glucosides) are a promising scaffold. However, because the entire manuscript does not analyze the actual activity of PLA2, it is evident that the statement in the manuscript: "PLA2 activity is inhibited and the conclusion that BBE can inhibit PLA2 oxidative stress," is at the risk of over-interpreting the data. So please be careful in describing the relationship between BBE-AgNPs and PLA2. In short, although idTarget can simulate the binding affinity score of the tested molecule to a specific protein, further activity analysis of this specific protein (PLA2) and the 13 metabolites found in the manuscript is still required to make a preliminary judgment about whether BBE has an impact The ability of PLA2 activity. Therefore, adding a note at the appropriate place in the text is recommended to explain the possible gap risk between the simulated data using idTarget software and the actual enzyme activity in the test tube.

3. The author should explain why 200 BBE-AgNPs are less effective than 50 BBE-AgNPs in Figure 4 and Figure 5.

4. The methodology of Entrapment efficiency is not transparent. Metal nanoparticles have a particular collective dipole oscillation characteristic of surface plasmon resonance (SPR) [doi:10.1021/CR900137K], and pH can have a significant impact on AgNPs by changing their surface charge and indirectly affecting particle stability and aggregation [doi:10.2478/S13536-013-0166-9, doi:10.1016/J.CHEMOSPHERE.2018.10.122]. It is unclear how different concentrations of BBE will affect these light absorption characteristics and even cause quantitative and BBE activity.

5. Some documents (Reference 44) have improper citations (Lines 194-198). Please read the full text carefully.

Minor suggestions:

1. In Figure 4 and Figure 5, should the arrow (Λ) be a significant difference symbol compared with Sham? Please mark it clearly in the figure description.

2. In Figure 5, why not measure GSH/GSSG?

3. Please mark the microscope's magnification on the IHC pictures; the magnification of some pictures seems different, for example, (B) A and S of Casp-9. Please confirm. In addition, Figure 9 (A) Bax and (B) Casp-9 treatment group coding are scattered and illogical.

Reviewer 2 Report

In this manuscript, “Blackberry-loaded AgNPs Attenuate Hepatic Ischemia/Reperfusion Injury via PI3K/Akt/mTOR Pathway” by Fathi et al. reports that BBE-AgNPs can protect against mitochondrial apoptosis caused by liver I/R which make BBE-AgNPs to be a potential therapeutic candidate. This work is well designed and written; therefore, I would suggest minor revision before publication. Here are the comments and suggestions:

1.     There are some typos on page 19.

2.     Section for Conclusions can be added.

3.     The low dosage of BBE seems better than that of high dosage. Could authors provide more results with various BBE concentrations?

Reviewer 3 Report

I read the manuscript entitled “Blackberry-loaded AgNPs Attenuate Hepatic Ischemia 2 /Reperfusion Injury via PI3K/Akt/mTOR Pathway” and have the following suggestions and comments to improve the state of the study.

Line 186: The authors should rewrite the following text: 'Wavelength and absorbance of prepared metal nanoparticles'.

The authors should discuss in the text Tables S1-S8 from the Supplementary Material (from non published Material).

The authors should insert a Conclusions section that is supported by more data results.

Minor points: Line 29: with their anti-inflammatory = for their anti-inflammatory; Line 188: ranged from 250 to 700 = ranging from 250 to 700; Line 326: as a major compounds = as major compounds; Line 327: have bben= have been; Line 511: insdie = inside.
